# Motivations for sustained use of modern contraceptives in a peri-urban area: An analytical cross-sectional study

Seth Amponsah-Tabi[1,2,3]*, Hannah Boatemaa Asante[4], Edward T. Dassah[5,6], Eric Sarpong Ansong[7], Maxwell Kankam[2,6], Amponsah Peprah[3,6], Charles Senaya[6], Jude John Annan[3,6], Roderick Larsen Reindorff[3,6], Henry S. Opare-Addo[3,6]

1 Department of Obstetrics and Gynaecology, Suntreso Government Hospital, Ashanti, Ghana, 2 Saint Patrick's Hospital, Offinso Municipal Area, Ashanti, Ghana, 3 Department of Obstetrics and Gynaecology, School of Medicine, Kwame Nkrumah University of Science and Technology, Ghana, 4 Sanford Health, Ghana, 5 School of Public Health, Kwame Nkrumah University of Science and Technology, Kumasi, Ghana, 6 Directorate of Obstetrics and Gynaecology, Komfo Anokye teaching Hospital, Ashanti, Ghana, 7 Insurance Quantified centre, Ghana

* sethonto@gmail.com

## Abstract

### Background

Discontinuation of contraceptives is one of the most significant concerns for family planning programmes. It is estimated that 1 in 3 women who take up modern contraceptives discontinue use by the end of a year due to various reported challenges. This study aimed to determine proportion of sustained users (continual use of the same method for 2 or more years) and their motivation.

### Methodology

A cross-sectional survey with a questionnaire was conducted among 397 women of reproductive age (15–49 years) in the Mfantseman municipality to assess the factors that motivate them to sustain the use of modern contraceptives. A multi-stage sampling technique was used to obtain the desired study participants.

### Results

39.8% of study participants had used their method of contraception for 2 years with 14.4% having used theirs for 4 or more years. Ever missing one's dose of their method, birth goals and the need for privacy when using a method had the highest odds for sustained use with aORs of 3.46, 2.53 and 2.09 respectively. About 40% of the respondents reported socio-economic reasons as motivation for sustained use.

**Data availability statement:** All relevant data are within the paper and its Supporting Information files.

**Funding:** The author(s) received no specific funding for this work.

**Competing interests:** The authors have declared that no competing interests exist.

## Conclusion

Findings from the study suggest it would be beneficial for service providers to demonstrate tolerance towards women who miss their doses or schedule of their chosen method of modern contraceptives. The intention for adopting modern contraceptive use of more than a third of the respondents was to improve their socioeconomic status.

## Introduction

Discontinuation of contraceptive use has progressively become one of the most significant challenges for family planning programmes and remains an area of concern due to its far-reaching influence on contraceptive use [1]. It brings about the 'leaking bucket phenomenon' [2], whereby programmes that aim at increasing the prevalence of contraceptive use are hindered by contraceptive discontinuation. Contraceptive discontinuation also leads to unintended pregnancies and unsafe abortion among women [3] which are part of the most troubling public health challenges and a major worldwide reproductive health issue that imposes an appreciable socioeconomic burden on individuals and society [4].

Family planning studies initially explored the determinants of contraceptive uptake, however there is an appreciable shift in focus of research from uptake to the continuation of use. Studies now suggest that continuity of contraceptive use is becoming an important measure for programme effectiveness as initial uptake increases. Jain asserts that family planning programmes would do better if the focus is shifted to satisfying and retaining existing users rather than on recruiting new clients [2,5]. Modern contraceptive use in Ghana is low as in many other developing countries. Among married women, for instance, only about 25% of them were reported to be using a modern method of contraception according to the Ghana Maternal health survey report [6]. This is relatively low and requires the need to scale up uptake among women. However, focusing on retaining current users could prove beneficial as sustained users could become promoters of contraceptive use.

It is established that there is a strong association between contraceptive use and reduced maternal and infant mortality, along with improved socioeconomic opportunities [7–10], hence there is the need to identify measures to eliminate discontinuation and increase sustained (use of same contraceptive method for 2 or more years) use of modern contraceptives in the population.

## Methodology

### Study design

The research was an analytical cross-sectional study aimed at investigating the motivations for modern contraceptive use and the relationship between these motivating factors and the sustained use thereof of modern contraceptives. The municipal had its 6 sub- districts recording a total of 1665 new acceptors of injectable and oral contraceptives and 3748 continuing acceptors of these same contraceptive methods for

the year 2021. New users were those women who registered in the year, adopting the use of injectable or oral contraceptives for the first time while continuing acceptors were those who were registered for continuing use in the following year. Continuing acceptors were registered for the new year irrespective of the time of registration in the previous year.

Out of the two totals above, the Mankessim sub-district had the highest proportion of new acceptors and continuing acceptors, contributing 27% and 32% of these totals respectively. The Anomabo sub-district had 24% and 25% respectively and the Saltpond sub-district followed with 20% of new acceptors and 19% of continuing acceptors. These 3 sub-districts contributed 71% of the total new users and 76% of continuing users. Therefore, subjects were chosen from these sub- districts to represent the municipality.

## Operational definitions

Sustained use of contraceptive implies the use of the same modern contraceptive method (oral pills and injectables) for 2 or more years.

Motivations for sustained use are the clinical and non-clinical factors influencing the use of the same modern contraceptive for 2 or more years.

New users were those women who registered in the year, adopting the use of injectable or oral contraceptives for the first time.

Continuing acceptors were those who were registered for continue use of the same modern method in the following year.

## Study Profile

The Saltpond Municipality, is one of the 22 districts in the Central Region. The district is bounded to the east by Gomoa East and West; to the west by Abura-Asebu-Kwamankese district, the south by Gulf of Guinea and the north by Assin and Ajumako-Enyan-Essiam Districts. The municipal capital is Saltpond with some major communities as Mankessim, Essarkyir, Dominase, Anomabu, Kormantse, Abandze, Otuam, Narkwa and Yamoransa. The health directorate has 6 sub-districts namely Saltpond, Mankessim, Anomabo, Abandze-Kormantse, Biriwa and Dominase. There are 32 health facilities including hospitals, health centres, clinics and CHPS zones in the municipality. There is a total population of 42,191 for women of reproductive age within the municipality.

## Sample size

The sample size was generated by the Cochran formula assuming a normally distributed population of contraceptive users in a relatively large population. A margin of error of 5%, a confidence level of 95% and a standard deviation of 0.5 were used. This conservative method with a standard deviation of 0.5 was used as it gives a large sample size and the proportion of women who had sustained use of modern contraceptives within the population was unknown.

Sample size = [(z score)^2 * (standard deviation) * (1-standard deviation)]/ (margin of error)^2 (z score of 95% confidence interval is 1.96).

Sample size = [(1.96)^2 * (0.5) * (1-0.5)]/ (0.05)^2

Sample size = 384.16 which was about 385. Catering for non-response rate gave us an approximate sample size of 397 study participants.

## Inclusion and exclusion criteria

The study included women in their reproductive period accessing injectable and oral contraceptive use for at least 12 months at the Mfantseman Municipality. Excluded from the study are women accessing other contraceptives aside the

oral and injectable hormonals. Other women who were excluded: were women outside the reproductive period; those that did not give consent for the study; those who missed their dose for a month or more; and those using the hormonal contraceptives for other medical purposes aside family planning.

## Sampling

The quota, simple and systematic random sampling methods were used in obtaining 397 eligible participants to voluntarily partake in the study. The quota sampling was employed at the initial stage to achieve a spread of the target population based on their contribution to the sum of the 2848 continuing acceptors of the selected contraceptive methods, within the 3 selected sub- districts. The 3 selected sub-districts, Mankessim, Anomabo and Saltpond, contributed 42%, 33% and 25% of this total respectively and hence that was the quota apportioned to each sub-district. Based on this proportion, 167, 131 and 100 women from each selected sub-district respectively, who suited the inclusion criteria for the survey were selected to participate in the study. Based on the sampling frame and the number of participants to be interviewed over the study period, we calculated a sampling interval whereby a every third client was interviewed after meeting the inclusion criteria. The first participant for each day was selected by simple random sampling. At these 3 selected places, women reporting for their dose of modern contraceptive for the day were approached randomly after they had received care and informed of the study. If they expressed interest and fitted the selection criteria, they were given a questionnaire to answer.

## Data collection

Data was gathered with the use of an open ended and closed questionnaire. The researchers conducted face-to-face interviews for all study participants. Data was collected over a period of 4 weeks (5th January to 31st January, 2022) from individuals accessing contraceptive services from selected facilities in the municipality. The family planning record cards of these individuals were assessed to confirm the duration of use of their contraceptives. Data on socio-demography, the perceived quality of care received at family planning units, challenges experienced with the use of their method of modern contraceptives and the motivations behind their adoption of contraceptive use were gathered from respondents. The period of use of modern contraceptives was counted in complete years and hence additional months were rounded down to the nearest year.

Survey questions were in English and were only translated for those with language barriers in this regard by the attendant. Data were collected concurrently at the selected centres. Before the actual study was conducted, the survey questions were tested with 20 respondents in the Cape Coast Metropolis to check for ambiguity and necessary reviews made. Data was intended to be collected online by Google forms but there were internet connectivity challenges and as such most were collected via printouts which were later entered unto Google forms. Subsequently, there was cleaning of duplicate information; inaccurate and incomplete data was sorted out and missing data were replaced.

## Data analysis

All graphs and models were generated in Python. Initial exploratory analysis of the responses from the questionnaires was done and data cleaning was made on some of the response columns.

The percentages of sociodemographic variables among participants were calculated and the prevalence of sustained use of modern contraceptives among the different categories was compared. Women who had used their method for 2 or more complete years were classified as sustained users whereas those who had used theirs for 1 year were categorized as new users. The prevalence of sustained use of modern contraceptives within the municipality was estimated based on this classification.

A logistic regression model was fitted and the response variable (continuous usage) was treated as categorical. Crude odds ratios were computed for selected variables and the significant variables adjusted for. The output coefficients were log odds, and were converted to odds. Characteristic variables with p-values <10% at the crude analysis were selected as

significant for discussion. Observations were classified into individuals who had continuously used their method of contraception for 2 or more years and otherwise. Sustained users were those who had used their method for at least 2 complete years, given that about 60% of those who first enroll in contraceptive use stopped within the first 36 months (Ali et al., 2012). Birth spacing of 2 years is also globally recommended for women after delivery of a live birth. Factors affecting sustained usage of contraceptives and their relationship between this dependent variable (sustained usage) and the independent variables. Characteristic variables with p-values <10% at the crude analysis stage were selected as significant for discussion. The significant variables were then analyzed for adjusted odds ratios at a significance level of 5%. For related variables, only one of them was selected. For instance, marital status 'Married' and 'Single' were related and both were significant, hence 'Single' was selected for adjusted analysis.

### Ethical consideration

Approval from the Committee on Human Research Publication and Ethics (CHRPE/AP/619/21) from KNUST was sought to conduct the study. The purpose of the study was duly explained to eligible respondents to make an informed decision of participation. Participants were made aware that they could opt out of the study any time they wished to do so without being victimized. Participants were assured of the confidentiality of their identity and information shared. As such, all participants were assigned codes of reference and their documents were not made available to persons not involved in the study.

Each study participant gave a written informed consent before being enrolled into the study. A participant leaflet information was explained and made available to participants. Each participant before the interview had to fill the participant information leaflet. Written consent was obtained from parents and guardians of study participants who were minors before they were included in the study. Both the minors and parents were well informed about the study before obtaining their consent for the study.

Approval was also sought from the municipal health directorate before conducting the study.

## Results

### Socio-demography and characteristics of respondents in the study

There was a total of 397 respondents in the study and their information gathered was categorized into 5 sections- socio-demography, current modern contraceptive use information, quality of care received for modern contraceptive use, challenges with the use of method of modern contraceptive, and motivations for the use of modern contraceptives.

The majority of the respondents, (more than 50%) had at least basic education, about 30% had SHS education with about 10% having tertiary education and just a few having no formal education (Table 1). Most of the respondents were traders with a few doing white-collar jobs. Partners of most participants at least had basic education. The education level of partners had a similar structure as that of the participants themselves. Most respondents' partners were artisans (about 28%), about 17% drivers and 13% traders. Again, the majority of them were married, accounting for about 60% of the participants with the rest either single or cohabiting. More than 80% of the respondents were Christians and the rest were Muslims.

About a third of the study participants had a reproductive goal of limiting childbirth with the larger remainder only seeking to avoid pregnancy for at least a year. The majority of the respondents had between 2 and 4 children (Table 1).

There was a normal age distribution curve among the respondents with the majority of the women aged between 20 and 35 years with a mean of 29.7128 years and a median of 29 years.

### Prevalence of sustained use of modern contraceptives in the district

Of the total respondents, 128 (32.2%) of them had used modern contraceptives (oral pills and hormonal injectables) for a year at the time of the study (Fig 1). Another 39.8% had used it for at least 2 years. About 14.4% had used modern contraceptives for 4 or more years. All participants had used their method of modern contraceptives for at least one complete

**Table 1. Socio-demographic data of respondents.**

| Sociodemography | Response variable | Percentage (%) |
|---|---|---|
| *Age group* | < 20 | 6.5 |
| | 20-24 | 18.4 |
| | 25-29 | 28 |
| | 30-34 | 21.7 |
| | 35-39 | 15.1 |
| | 40-44 | 8.3 |
| | 45-49 | 2 |
| *Occupation* | Trading | 45.3 |
| | Teaching | 10.8 |
| | Unemployed | 9.6 |
| | Artisanship | 9.3 |
| | Schooling/ apprenticeship | 7.8 |
| *Highest education level* | Nil | 3.8 |
| | Basic | 53.7 |
| | SHS | 31 |
| | Tertiary | 11.8 |
| *Religion* | Christian | 88.4 |
| | Muslim | 11.3 |
| | Non-religious | 0.3 |
| *Number of children* | 1 | 25.9 |
| | 2 | 20.4 |
| | 3 | 20.7 |
| | 4 | 10.3 |
| | 5 or more | 10.6 |
| *Marital status* | Single | 19.1 |
| | Married | 62.7 |
| | Cohabiting | 18.1 |
| *Partner's highest education level* | Nil | 2.3 |
| | Basic | 61.0 |
| | SHS | 25.4 |
| | Tertiary | 11.3 |
| *Partner's occupation* | Driving | 16.6 |
| | Artisanship | 28.5 |
| | Teaching | 9.6 |
| | Fishing | 9.1 |
| | Schooling/ apprenticeship | 2.3 |

year from their start date. Individuals who had used the same contraceptive method for 2 or more years were classified as sustained users with about 68% in this category. With a 95% confidence interval, the population proportion of sustained users of modern contraceptives in the district was estimated between 0.6317 and 0.7243.

**Current modern contraceptive use information**

The majority of the study participants had used their method of contraception for 2 years (39.8%) with only 14.4% having used theirs for 4 or more years. Most respondents (68.5%) were using modern contraceptives to space childbirth. 342

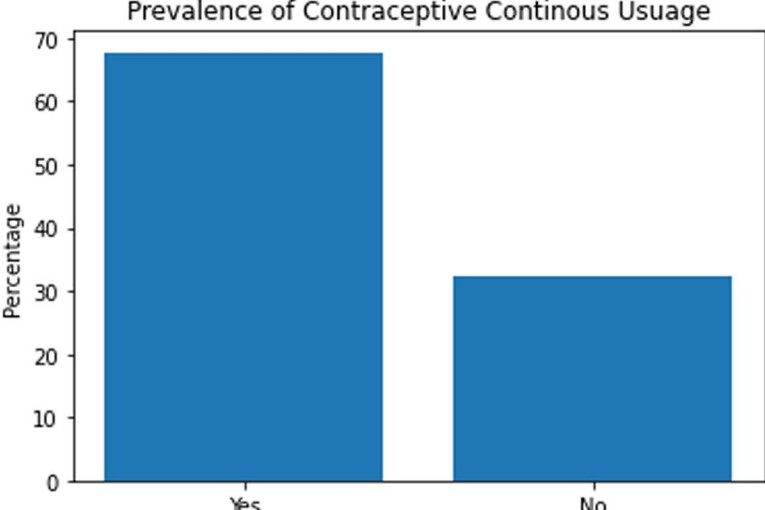

**Fig 1. Proportion of sustained (continuous) use of modern contraceptives.** The graph shows that about 68% of participants have used the pills and injectables for at least 2 years.

(86.4%) of the respondents used injectables while the rest used pills. Almost half of the total respondents were using their current method on the recommendation of associates or simply because their associates were also using them. About a third of the participants chose theirs on account of privacy. About 80% of the subjects reported that they had used a different type of modern contraceptive previously before switching to their current method and the reason for their switch was mainly due to the side effects they experienced while on the previous method (Table 2).

## Quality of care received by sustained users

The quality of care received by respondents was generally perceived as adequate by both new and sustained users. Almost all respondents indicated that they were educated on the process of menstruation and informed of the various available contraceptive methods to choose from. They also reported being educated about how their chosen method works to prevent pregnancy and the possible side effects to expect (Table 3). Almost all study participants interviewed were of the view that providers were friendly, interactive and permitted them to ask questions.

## Challenges experienced by sustained users

The majority of the study participants reported having challenges using their current chosen method of modern contraceptives. Generally, their challenges did not make them want to discontinue the use of their method. Some of the challenges included side effects, unfavourable schedules, partner disapproval, poor quality of care and inadequate privacy. Side effects of the method used were the most common challenges reported by about 85% of the respondents, with the dominant effect being irregular or unscheduled bleeding (about 80%). Other side effects reported included headache, weight gain, weight loss, dizziness and low libido.

Most respondents reporting side effects as a challenge indicated they did not report to their facilities of care for management. About 35% of them indicated that they did not do so as they were pre-informed. About 32% reported it was because they realized they just had to adjust and cope with those effects and about 20% stated that the effects were not of health concern to them. The remaining respondents reported reasons such as long waiting time at the facility. For those that reported their side effects for management, more than half reported that it did not relieve the effects.

**Table 2.  Current modern contraceptive use information.**

| Current use information | Response variable | Response (%) |
|---|---|---|
| *Duration of use (years)* | 1 | 32.2 |
| | 2 | 39.8 |
| | 3 | 13.6 |
| | 4 or more | 14.4 |
| *Childbearing goal* | Space childbirth | 68.5 |
| | Limit childbirth | 31.5 |
| *Current method of contraceptive* | Injectables | 86.4 |
| | Pills | 13.6 |
| *Reason for current method* | Privacy | 37.8 |
| | Less side effects | 10.0 |
| | Favourable schedule | 2.8 |
| | Recommendation | 49.1 |
| | Other | 0.8 |
| *Use of a previous method* | Yes | 80.1 |
| | No | 19.9 |
| *Reason for switching* | Privacy | 12.5 |
| | Side effects | 65.0 |
| | Favourable schedule | 5.0 |
| | Other | 17.5 |

**Table 3.  Quality of care received for modern contraceptive use.**

| Quality indicator | Response variable | Total response (%) | New users response (%) | Sustained users response (%) |
|---|---|---|---|---|
| Educated on process of menstruation | Yes | 94.7 | 94.5 | 94.8 |
| | No | 5.3 | 5.5 | 5.2 |
| Informed of various contraceptive methods | Yes | 99.2 | 100.0 | 98.9 |
| | No | 0.8 | 0.0 | 1.1 |
| Educated on how chosen method works | Yes | 93.0 | 93.8 | 92.6 |
| | No | 7.0 | 6.3 | 7.4 |
| Educated on possible side effects | Yes | 99.0 | 98.4 | 99.3 |
| | No | 1.0 | 1.6 | 0.7 |
| Given the chance to ask questions | Yes | 98.0 | 96.9 | 98.5 |
| | No | 2.0 | 3.1 | 1.5 |
| Received adequate information | Yes | 96.0 | 93.8 | 97.0 |
| | No | 4.0 | 6.2 | 3.0 |
| Met friendly and interactive provider | Yes | 97.5 | 96.1 | 98.1 |
| | No | 2.5 | 3.9 | 2.9 |
| Seen by same provider each time | Yes | 23.2 | 26.7 | 21.6 |
| | No | 76.8 | 73.3 | 78.4 |

Most of the respondents had heard negative rumors (misconceptions) about modern contraceptives but that generally did not inspire discontinuation. Surprisingly, about 2% of the total respondents reported they had not heard any misconceptions about modern contraceptives (Table 4).

**Table 4. Challenges with the use of chosen modern contraceptive.**

| Challenges | Response variable | Total response (%) | New users' response (%) | Sustained users' response (%) |
|---|---|---|---|---|
| Any challenges with use of method of contraception? | Yes | 71.5 | 70.3 | 72.1 |
| | No | 28.5 | 29.7 | 27.9 |
| Do challenges make you want to stop using modern contraceptives? | Yes | 26.5 | 32.6 | 23.7 |
| | No | 73.5 | 67.4 | 76.3 |
| Side effects reported for management? | Yes | 62.5 | 68.8 | 59.8 |
| | No | 37.5 | 31.2 | 40.2 |
| Does management relieve side effects? | Yes | 45.2 | 44.4 | 45.6 |
| | No | 54.8 | 55.6 | 54.4 |
| Heard of any rumours about modern contraceptives? | Yes | 97.7 | 97.7 | 97.8 |
| | No | 2.3 | 2.3 | 2.2 |
| Do rumours make you want to stop using modern contraceptives? | Yes | 19.6 | 19.1 | 19.8 |
| | No | 80.4 | 80.9 | 80.2 |
| Ever missed a dose/schedule of your method? | Yes | 38.5 | 22.7 | 46.1 |
| | No | 61.5 | 77.3 | 53.9 |

About 38% of the total respondents reported ever missing a dose of their chosen method that made them think they were at risk of pregnancy; however, they took measures to prevent pregnancy. More sustained users (46.1%) than new users (22.7%) reported missing a dose. Among sustained users, about 60% avoided sexual intercourse; about 30% resorted to emergency pills and about 10% used condoms to prevent pregnancy in the event of missing a dose (Figs 2–6).

## Motivations for sustained use of modern contraceptives

About half of the total study participants indicated that they knew a healthcare provider who used modern contraceptives. Some of these women had had their providers share their experiences with them while most participants reported that their providers did not reveal their status to them on their initial visits. This was a similar trend among both new and sustained users. Almost all respondents had close ties using modern contraceptives who had shared their experiences with them. Again, almost all sustained users indicated that the reasons for initially adopting modern contraceptive use motivate them to continue use while a very few of them (0.4%) had other reasons for sustained usage. Generally, all respondents believed the benefits of modern contraceptive use outweighed the challenges with use.

Several reasons motivated continuous use among sustained users which included limiting childbirth, saving more money, spacing births, and granting partners' requests among others. More than 30% of the sustained users reported limiting childbirth as their motivation. More than a quarter of them indicated it was to save more money before giving birth ever or having their next child. For about 15% of them, it was to space their birth while about 10% wanted to ascertain their union with their partners before childbirth. About 8% indicated they wanted to finish schooling/apprenticeship; about 5% stated it was to get a job and the rest reported reasons as simply delaying pregnancy, granting partners' requests and other undefined reasons (Table 5).

## Relationships between observations and sustained use of modern contraception

The study revealed that age, number of children, birth goals, switching from a different contraceptive method, privacy as a reason for choosing the current method, individuals with tertiary education, individuals who are happy with their providers and ever missing a dose or schedule of the current method were all statistically significant to sustained use of modern contraceptives. Their p-values were all less than the significance level of 0.05. The following variables- number of children, switching from a different contraceptive, tertiary education and being happy with providers- had a negative

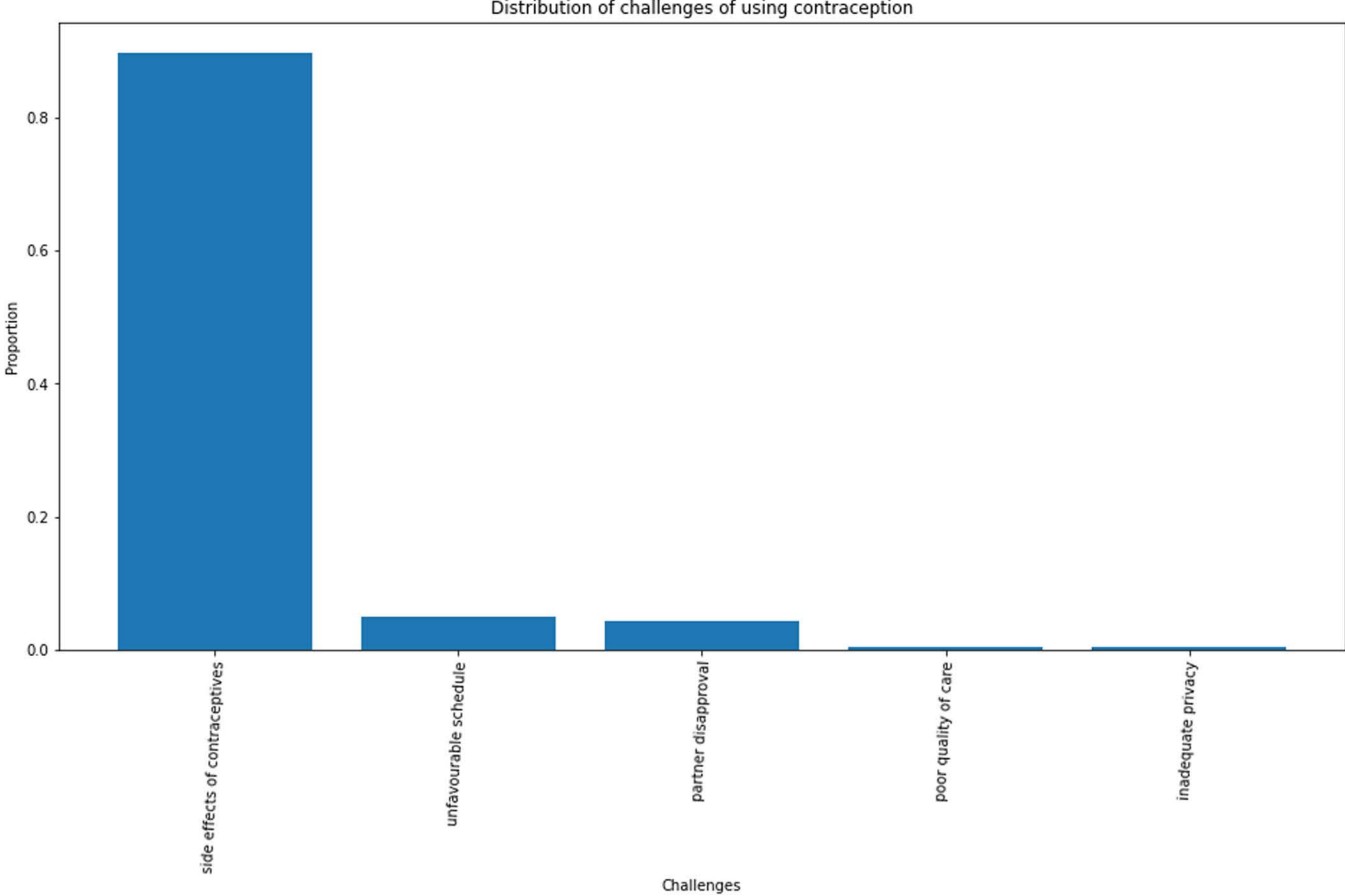

**Fig 2. Distribution of challenges with usage of modern contraceptive.** This graph shows that majority of challenges to sustained use is from side effects of the methods. This is followed by unfavourable schedules to the family planning unit and partners interferences with the woman's reproductive choices.

relationship with continuous usage while the remaining variables were otherwise. Those who were using a different modern contraceptive before switching to the current method were less likely to continue usage. Likewise, people with tertiary education; those with more children and those who were happy with their providers were less likely to sustain the use of modern contraceptives, holding other factors constant. These variables had an odds ratio of less than 1.

Conversely, age was a significant correlate of sustained contraceptive use {95% CI 1.1, 1.2 p value 0.000}. Similarly, individuals who reported privacy as their reason for choosing their current method of contraception were about twice more likely to have sustained use 2.09 {95% CI 1.24, 3.53 p value 0.005}. Study participants whose childbearing goal was to limit child-birth were about 2.5times more likely to have sustained use than spacers {95% CI 1.05, 6.10. p value 0.038} and individuals who had ever missed a dose or a schedule of their method were 3.46 times {95% CI 2.02, 5.96 p value 0.000} more likely to have sustained use of modern contraceptives, holding other factors constant (Table 6).

## Discussion

### Socio-demography of respondents

The relationship between religion, education, partner's education or occupation of respondents and sustained use of modern contraceptives did not have any statistical significance. This is quite at variance with other studies. For instance,

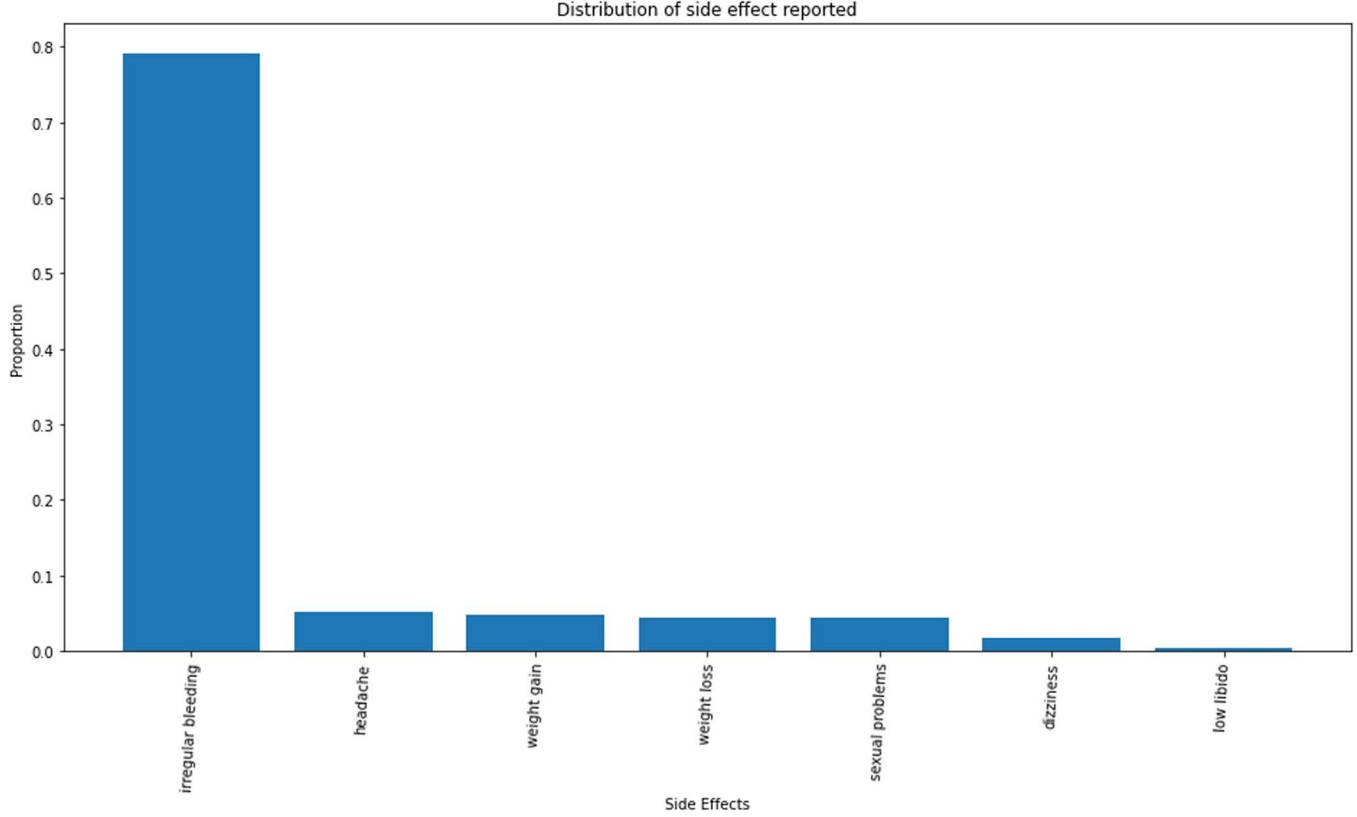

**Fig 3. Distribution of the side effects.** This graph shows irregular bleeding per vaginam as number one complaint. Headaches and weight changes were also reported.

Alvergne et al. [11] found that religion could influence use of modern contraceptives as their study revealed abandonment of contraceptive use was higher among Muslims than Christians. Also, discontinuation of contraceptive use has been reported to be higher among women in the poverty bracket or making lower income [11,12]. These discrepancies might be due to some contextual differences and the population that was studied.

Age, marital status, number of children and birth goals of respondents however, had a significant correlation with sustained use of modern contraceptives. Age has been found to influence sustained use of modern contraceptive by other studies. Silumbwe et al., [12] reported from their study that younger women are more likely to discontinue use of their modern contraceptive than older ones. The younger women may not have stable relationships and thus are more likely to discontinue use [13]. A low desire for fertility could translate to a high motivation to control fertility. This is known to increase the likelihood to continue use of contraceptives [14]. The age distribution among the respondents shows a normal curve with the majority of the women aged between 20 and 35 years. It is necessary for women in the prime of their fertility period to lower their number of pregnancies and also to space them adequately to decrease their risk for maternal morbidities and or even mortalities. This is highly crucial because in the absence of pregnancy, the risk of maternal death is non-existent. This necessitates sustained use of modern contraceptives among such women as they are established to be more effective in preventing pregnancy.

### Prevalence of sustained use of modern contraceptives in the district

The majority of the women (86.4%) were using injectable method of modern contraceptives and not pills. This was expected as injectables, particularly the Medroxyprogesterone injection (Depo Provera), are noted to be the most popular

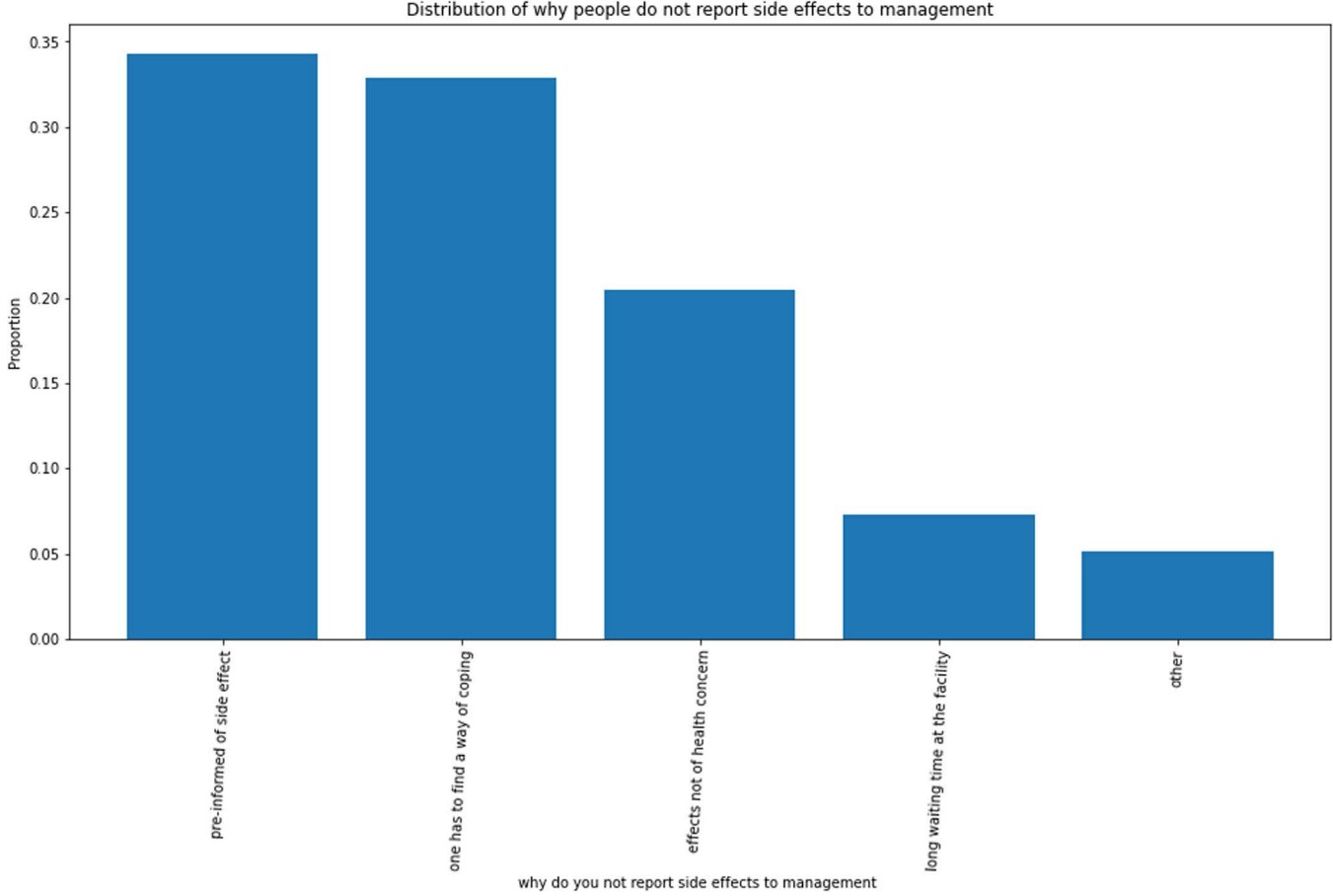

**Fig 4. Reasons for not reporting side effects.** The majority of those who did not report side effects took the decision because they were pre-informed during counseling before uptake. The others decided to cope with the side effects while others did not consider it as a significant health concern.

modern contraceptives among users in sub-Saharan Africa as reported by many surveys [6]. Studies from other areas like Ethiopia have reported similar findings where about 65% of their study subjects were taking Depo Provera shots [15]. The 2017 Ghana Maternal Health survey showed the injectables (especially Depo provera) as the most patronized modern contraceptive among both married and unmarried women [6].

Of the total respondents, 158 of them (39.8%), had used modern contraceptives for 2 years at the time of the study while 13.6% and 14.4% had used it for 3 years and 4 or more years respectively. All participants had continued use of their method of contraceptives for at least one complete year from their start date and the sustained users, as defined by this study, accounted for 69% of the study participants. This is surprising as studies have found that about a third of initial adopters stop at the end of a year and yet about half stop by the second year [9]. It is noteworthy that some participants reported having switched from one modern contraceptive to their current method. However, the study was exclusive to the absolute duration in years of their current method of contraceptive and did not capture many details concerning such an event. This suggests the prevalence of continuous use of modern contraceptives may be underestimated without consideration to the interval between switching and the reason for doing so.

Also, about a half of the sustained users reported ever missing a dose but took measures to prevent pregnancy which included use of condoms and emergency pills. Although the exact period of missing them was undefined, in the study, the

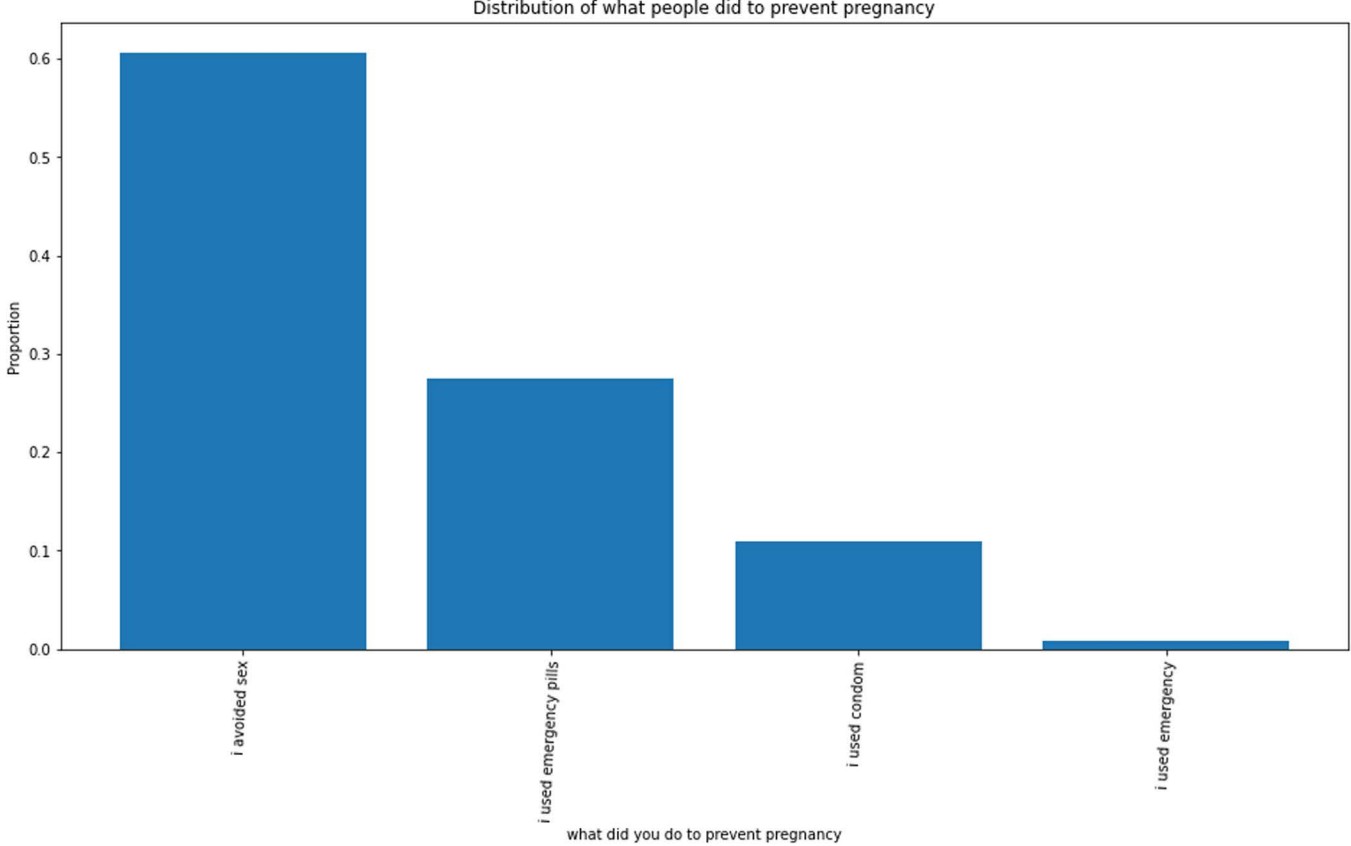

**Fig 5. Measures to prevent pregnancy after a missed dose.** The distribution shows that the majority preferred temporary abstinence after missing a dose. Other resorted to Emergency contraceptive pill and condom use.

objective of contraception was present. Moreover, emergency pills and condoms are also modern contraceptives. This illustrates that the women not reporting for their doses did not necessarily mean they had abandoned use of their method; counting such periods for a discontinuation event may not be justified. This conforms to the established fact that not all discontinuation event of modern contraceptive use is problematic [10].

## Quality of care received by sustained users of modern contraceptives

There was no significant difference noted between the quality of care as perceived by both sustained users and new users. Over 80% of all respondents affirmed they were educated on the mechanism of menstruation and pregnancy. Most of the respondents admitted that they were informed about the various methods of contraception available on their first visit. Again, more than 80% of all respondents indicated that they were educated on how their chosen method of modern contraceptive works to prevent pregnancy and the possible side effects were mentioned to them on their first visit. The majority of the women in this study, again, admitted they were allowed to ask questions and believed they were given adequate information about their method of choice. This result is impressive as it suggests the desire of women to receive comprehensive information about family planning as reported by Dehlendorf et al. [16] was generally met in the municipality.

Again, over 80% of the respondents indicated they were satisfied with their first visit and believed their providers were friendly and interactive. This is a reassuring discovery as studies show that women desire friendly relationships with

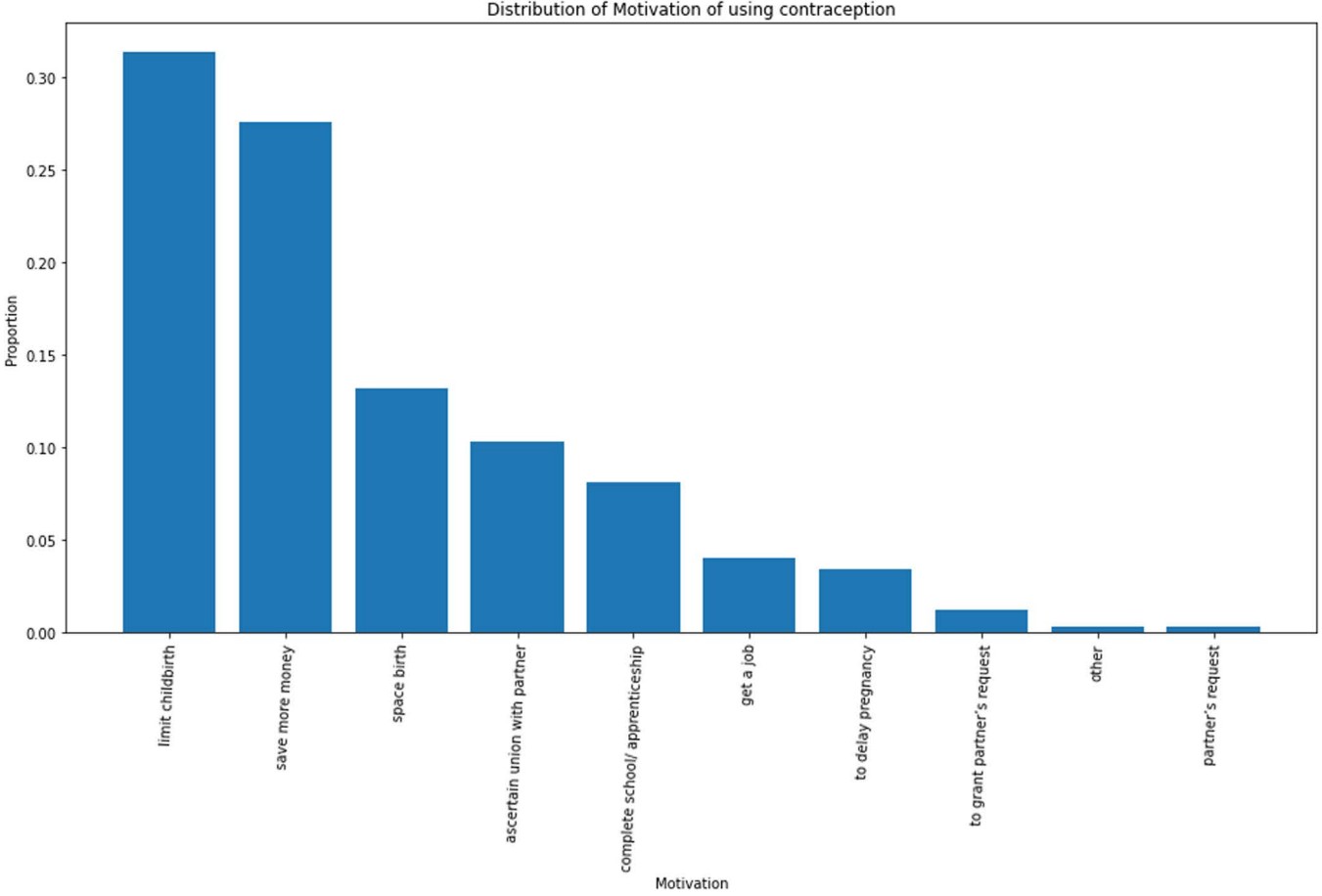

**Fig 6. Motivations for sustained contraceptive use.** The distribution shows that a woman's desire to limit child birth is the greatest motivation for sustained use followed by economic burden and spacers.

**Table 5. Motivations for the use of modern contraceptives.**

| Motivations | Response variable | Total response (%) | New users' response (%) | Sustained users' response (%) |
|---|---|---|---|---|
| Know healthcare provider using contraceptives | Yes | 49.1 | 50.8 | 48.3 |
| | No | 50.9 | 49.2 | 51.7 |
| Healthcare provider share their experience with you | Yes | 54.2 | 65.5 | 49.2 |
| | No | 45.8 | 34.5 | 50.8 |
| Provider revealed their status on first visit | Yes | 37.4 | 39.1 | 36.6 |
| | No | 62.6 | 60.9 | 63.4 |
| Have close ties using modern contraceptives | Yes | 97.5 | 96.8 | 98.5 |
| | No | 2.5 | 3.2 | 1.5 |
| Close ties share their experience with you | Yes | 97.9 | 96.8 | 98.5 |
| | No | 2.1 | 3.2 | 1.5 |
| Reasons for adopting contraceptive use as motivation for continuous use | Yes | 99.7 | 100.0 | 99.6 |
| | No | 0.3 | 0.0 | 0.4 |
| Contraceptives benefits outweighs the challenges | Yes | 97.2 | 96.1 | 97.8 |
| | No | 2.8 | 3.9 | 2.2 |

**Table 6. Relationship between observations and sustained use of modern contraceptives.**

| Characteristic Variables | cOR (95% CI) | P-value | aOR (95% CI) | P-value |
|---|---|---|---|---|
| Age | 1.10 (1.06, 1.14) | 0.0000 | 1.15 (1.1, 1.2) | 0.0000 |
| Number of children | 1.26 (1.09, 1.45) | 0.0016 | 0.67 (0.52, 0.88) | 0.0033 |
| Birth goals | 3.18 (1.87, 5.40) | 0.0000 | 2.53 (1.05, 6.10) | 0.0384 |
| Type of Contraceptive | 0.95 (0.52, 1.74) | 0.8644 | – | – |
| Switch from a method | 0.64 (0.38, 1.06) | 0.0840 | 0.41 (0.22, 0.75) | 0.0040 |
| Reason for current method (favorable schedule) | 1.28 (0.33, 4.92) | 0.7171 | – | – |
| Reason for current method (Less side effect) | 0.80 (0.40, 1.61) | 0.5317 | – | – |
| Reason for current method (privacy) | 1.45 (0.93, 2.27) | 0.0981 | 2.09 (1.24, 3.53) | 0.0055 |
| Reason for current method (other) | 0.95 (0.09, 10.63) | 0.9700 | – | – |
| Reason for current method (Recommendation) | 0.75 (0.49, 1.14) | 0.1767 | – | – |
| Basic Education | 1.64 (1.07, 2.49) | 0.0200 | – | – |
| SHS Education | 0.68 (0.43, 1.07) | 0.0900 | – | – |
| Tertiary education | 0.52 (0.28, 0.98) | 0.0420 | 0.44 (0.21, 0.92) | 0.0291 |
| Happy with provider | 0.58 (0.30, 1.11) | 0.1000 | 0.46 (0.21, 0.99) | 0.0473 |
| Providers status revealed | 0.90 (0.58, 1.39) | 0.6312 | – | – |
| Missed a dose | 2.94 (1.82, 4.74) | 0.0000 | 3.46 (2.02, 5.96) | 0.0000 |
| Able to express concerns | 2.13 (0.52, 8.65) | 0.2909 | – | – |
| Cohabiting | 0.88 (0.51, 1.50) | 0.63 | – | – |
| Married | 1.64 (1.07, 2.52) | 0.02 | – | – |
| Single | 0.55 (0.34, 0.92) | 0.02 | 0.74 (0.37, 1.50) | 0.4099 |

their providers. Fruhauf et al., [14] found from their study that such positive client-provider interactions and high satisfaction of clients during their first encounters, sequentially have a useful influence on their continuous use of modern contraceptives.

Almost all respondents reported having a generally positive experience during their visits and yet more than 80% of them expressed the desire to meet the same provider on each visit. This may be due to friendship and trust built with particular providers or rather the desire to have such relationships built over time. This agrees with study findings that women desire intimate, friend-like relationships with their providers [16]. This demand however could not be satisfied as the majority of the participants indicated they did not meet the same service provider on each visit due to reasons such as the rotation and shift model of our healthcare system and relocation by clients.

## Challenges experienced by sustained users of contraceptives

About 70% of all respondents in the study admitted to having challenges using their method of contraception, with only 30% of them reporting possible discontinuation of use due to these challenges. Both new and sustained users were similarly confronted by challenges, as about 70% of either category reported such circumstances. Nonetheless, only about 20% of the sustained users with challenges had thoughts of discontinuation while almost 40% of the new users had such consideration. It is worth noting that all respondents were still using their method at the time of the study and were recruited only after receiving a dose of their method. For these individuals to have considered discontinuation and yet report for their dose, connotes some positive implications. Firstly, this implies the existence of "unmet need" in the context of contraceptive discontinuation and this gap could provide an opportunity for providers to intercept users' decision to discontinue as this period of tentativeness may be long enough to take action. Secondly, this suggests there could be factors strong enough to have overcome the desire of discontinuation of modern contraceptive use to eschew its related challenges. Such factors, when carefully explored, could help both

policymakers and healthcare providers tailor the provision of care to women to promote the sustained use of modern contraceptives.

Challenges reported included partner disapproval and unfavourable schedule for dose intake among others. However, the most dominant of them was the experience of side effects which was reported by 80% of sustained users with challenges. This coincides with findings from other studies like that of Bellizzi et al., [17]. This might have had the highest frequency among the others because it is acknowledged that side effects are the most problematic issue with use of modern contraceptives for women [18,19,20]. The principal side effect reported was irregular bleeding aside others as weight gain, weight loss, and sexual dysfunction among others. This indicates that it would be presumptuous to conclude that all women who continue the use of modern contraceptives do so because they have no challenges. Only about 30% of the women in the study did not report back to their health facility for management of side effects though management was nevertheless seemingly unhelpful to most (70%) of those who reported. New users were more likely to report side effects for management than sustained users as over 60% of the former, admitted reporting side effects. The chief reasons for failure to report for such management were foreknowledge of possible side effects (35%) and the notion that side effects were simply to be endured (30%).

It is interesting to note that about 30% of all respondents denied hearing any rumours and misconceptions about modern contraception. Of the remaining 70% who knew about such rumours and misconceptions, about 30% of them revealed that they ever stirred thoughts of discontinuation in them. This corresponds with the findings of Farmer et al. [21] as well as Yee and Simon [22]. This similarity is not surprising because as social beings, people everywhere could be influenced either positively or negatively by misconceptions and rumours in all matters, which includes modern contraception. This warrants the need for providers to foster trusting relationships with users and be sociable and agreeable within their communities of practice so they exert a positive influence on clients. Rumours and misconceptions did not significantly affect the continuous use of modern contraceptives in both new and sustained users. The remaining individuals who were not perturbed by such misconceptions and rumours, may not have been experiencing any side effects as about 20% reported not having any challenges or possibly were moved by some stronger factors or desires that fortified their decision to use modern contraceptives.

Interestingly, most respondents reported having challenges using their method of contraception but continued use. This may have been due to a strong desire to attain better socio-economic statuses as the reported reasons for adopting modern contraceptive use were essentially within the backdrop of socioeconomic advancement. Thus, this might have encouraged them to do some trade-offs for a substantive gain, which to them was worth enduring the challenges for. This seems to be consistent with findings of Keogh et al., [23]. They discovered from their research that although menstrual changes were the leading cause of some women discontinuing the use of hormonal modern contraceptives, many exhibited the willingness to endure these effects for their attribute of effectiveness in preventing pregnancies. This shows women could make tradeoffs where necessary and may be more willing to adjust and continue use of modern contraceptives than we anticipate.

About 40% of the total respondents admitted ever missing a dose of their chosen method that made them think they were at risk of pregnancy. Almost 50% of sustained users ever missed their dose compared to about 35% of new users who missed a dose. This is anticipated as compliance with medication intake for a longer duration is obviously difficult for clients. Of the women who had ever missed a dose, more than half revealed they abstained from sex and about 30% indicated they used emergency pills to prevent pregnancy during that period. This is highly impressive as it suggests these women had a strong desire to prevent pregnancy and did not underestimate the possibility of its occurrence. Their ability to take action to make up for their omission most likely gave them a sense of control, which influences people's ability to decide to engage in an act as stipulated by the theory of reasoned action known to guide traditional contraceptive behavior [24].

None of the respondents thought their reasons for adopting modern contraceptive use were unimportant with all respondents choosing 3 or 4 on a scale of 1–4 for the importance placed on these reasons. Most people would believe the

reasons for their actions are significant, and hence this was logical. Contraceptives purposefully prevent pregnancy hence benefits derived from their use are practically based on the interests (reasons) that the absence of pregnancy serves and their gain thereof. It is therefore coherent that majority of these women reckoned that the benefits of modern contraceptive use outweighs its challenges for them to have adopted their use despite possible challenges. However, some women, about 20%, held contradicting views in this regard. This possibly arose from respondents who reported not experiencing any challenges using modern contraceptives. Again, it could be from respondents who may have been coerced into using modern contraceptives by circumstances beyond their control. The latter possibility comports with findings of Benson et al., [25] who reported from their study that women who choose a contraceptive method based on their provider's method is likely to discontinue use. It was noted some women reported using modern contraceptives at their partner's request and some also on a physician's counsel after previous obstetric complications. Such circumstances may cause women not to appreciate or acknowledge the significance of modern contraceptive use.

## Motivations for sustained use of modern contraceptives among users

All the respondents had their peculiar intentions for their decision to contracept. More than 50% of the respondents chose modern contraceptives because they wanted to limit childbirth for optimum care of their existing children. Less than 20% reported their intent as the need to save money before having a child or their next child and about 10% indicated they wanted to space their births for adequate care of their children. The remaining respondents had reasons such as wanting to ascertain the union between them and their partners; completing school/apprenticeship and granting their partner's request. These were the intentions behind the reproductive (birth) goals of the women in this study- the purpose for which they decided to adopt the use of modern contraceptives. These primarily signified their desire to better their socio-economic circumstances. This implies that some of the women understood the socioeconomic importance of modern contraceptive use as similarly reported by Farmer et al [21] and Silumbwe et al [12]. They reported from their study, that women understand that family planning services help women concentrate on their work and other socioeconomic activi-ties. It has been established unintended pregnancies along with high fertility essentially lead to reduced quality of life and workforce efficiency which is a public health concern. Decreasing such incidences promote public health by decreasing youth dependency and increasing women in paid labour [4,8]. It is therefore reassuring to discover women are gradually appreciating the socio-economic importance of modern contraceptives and for that matter, their sustained use.

There were 2 birth goals defined in the study- 'to limit childbirth' or 'to avoid childbirth for at least a year'. The reported birth goals responses significantly influenced sustainability of the use of modern contraception. Those who had the birth goal of limiting childbirth were more likely to sustain the use of modern contraceptives and this agrees with the findings of OlaOlorun, et al. [26]. This is expected as those who want to limit childbirth are more likely to be older women whose desire or ability for birth naturally might have dwindled.

Most of the respondents did not know any healthcare providers that used modern contraceptives. Only 30% stated that their providers revealed their status of contraceptive use to them on their first visit. About 20% of respondents declined ever sharing their experiences of modern contraceptives use with any close ties who also used them. For some respon-dents, the knowledge of close ties and healthcare providers also using modern contraceptives and the sharing of experi-ences encouraged them to continue use. This again, conforms to expectations as social beings and the influence of social pressure on people. Farmer et al., [21] found that women, particularly adolescents and unmarried women, were affected by the stigma associated with the use of contraceptives based on cultural and religious beliefs. Benson et al., [25] showed in their study how women are influenced by social factors regarding the use of modern contraceptives. It was found that women were more likely to use a type of modern contraceptive if they knew their providers were using them and liked to share experiences with their associates. It is believed that when a provider reveals their status, a client may be subtly persuaded to choose the provider's method which may not be her choice and often, later, brings about dissatisfaction and discontinuation. Nevertheless, other studies have also found women that, though women want control over the ultimate

selection of their method, they want their providers to participate in deciding on the method they choose [16]. There is thus a need for providers to learn the skill of assisting women to choose their methods without overtly imposing their preferences on them.

## Relationships between selected factors and the sustained use of modern contraceptives

From the results, respondents' age, marital status, education, number of children, birth goals, switching from a different modern contraceptive, reasons for choosing the current method of contraceptive, having challenges with the use of their method and ever missing a dose of their method were all statistically significant in relation to sustained use of modern contraceptives, with a 5% significance level. The education status of the respondents had a negative relationship with sustained use of modern contraceptives. Those with tertiary education had odds of 0.44 for sustained use of modern contraceptives and this agrees with findings from the Ghana Maternal health survey report, [6]. Single women were more likely to not sustain use of modern contraceptives with of 0.74. This is again consistent with the Ghana Maternal health survey report, [6]. This phenomenon could be because single women may have periods of celibacy where contraceptives use become needless, leading to discontinuation of use.

There was a negative relationship between continuous usage of modern contraceptives and the number of children a respondent had. The lesser the number of children a woman had, the more likely she was to sustain the use of her method. This could be because the women have realized the need for socioeconomic independence similar to Silumbwe et al.'s findings. It appears more women are working to support the home in recent times. Most of the respondents (about 40%) were traders and given that they were mainly self-owned small-scale businesses, they possibly needed time to also focus on growing their businesses and not only on child- raising.

Individuals who had a friendly and interactive experience on their first visit were more likely to discontinue using modern contraceptives. This is a contradictory finding to other studies such as that of Silumbwe et al. [12] and may be due to a confounding factor within the study.

For statistically significant variables, the reason for choosing the current method (those who were scared of implants or had unpleasant experiences with implants) had the highest odds. This means that for individuals whose reason for choosing their current method of contraceptive was for being "scared of implants", the odds that they would continue to use (other) modern contraceptives were 2 times as large the odds that they would not, holding other factors constant. This indicates how strong the reason for a particular decision could motivate an individual to continue in a behaviour.

Also, for an individual whose birth goal was to limit childbirth, the odds she would continue to use modern contraceptives was about 2.6 times as large as she would not, holding other factors constant. This is possibly because women whose reproductive goal was to cease childbirth were more likely to take keen steps to prevent pregnancy as compared to those who desired to have more children at a later time- this comports with the findings of OlaOlorun et al [26]. It may be because the latter could be ambivalent about their decision or simply not be much perturbed about the occurrence of pregnancy while they wait. It is reported by several researchers including Speizer et al., [27] that there tend to be inconsistencies between women's fertility motivations and their contraceptive behaviours. This indicates how important it is for providers to review women's reproductive goals with them when they visit for family planning care as it would sequentially have a far-reaching influence on their ability to continue use against all odds.

In contrast, there was a negative relationship between sustained use and switching methods of modern contraceptives. This may be explained by the fact the likelihood of a re-occurrence of behaviour is higher if it has ever happened. It could be deduced that individuals who had ever discontinued the use of any method of contraception could easily consider discontinuation and eventually repeat such event.

Women who chose a particular method due to privacy were more likely to have sustained use of the method. Since there are male partners and other family members who disagree to contraceptive use by women in their reproductive age, privacy and confidentiality is very necessary for sustained use of contraception. These women (clients) in order to assert

their reproductive rights and still maintain marital and family harmony will want to sue contraception in privacy. A systematic review and meta-analysis in sub-Sahara Africa showed that privacy and confidentiality were significant correlates of contraceptive use [28].

Women who ever missed their dose had higher odds, 3.4 times more, to continue the use of their method than those who had not. It is striking that a phenomenon that appears inefficient had high odds for sustained use of modern contraceptives but it acknowledged that all discontinuation episodes are not necessarily problematic [10] and hence this may present opportunities for providers to better support clients in sustaining use of the modern contraceptives. The majority of the women who missed a dose seemed to know when they were in a precarious situation to take action to protect themselves against unintended pregnancies. Missing a dose appears to have given the women perceived sense of control by allowing them the autonomy to decide what substitute measures to execute for prevention of pregnancy. This finding is rewarding as it suggests that if women are supported to feel in control of their decisions, they are more likely to take measures to prevent unintended pregnancies which would include sustaining modern contraceptive use.

### Study Limitations

The period of continuous use was simplified and counted in complete years. This probably affected the proportion of sustained users as there were respondents who for instance could have been completing another full year within the next 3 months at the time of the study yet the months were rounded down to the nearest year.

Participants were required to give an account of the quality of care received, including that of their very first visit to the family planning unit. There is no universal scale for quality measurement. Quality assessment was based on client satisfaction and availability of certain standard services which in some other populations might be average or below average quality. Client's feeling of satisfaction might also be biased due to different encounters she might have had at the facility.

Some of the respondents had their initial contraceptive visit at different facilities in different districts or even regions which may be dissimilar in characteristics to the study area. This may have affected the quality-of-care account reported and may not be completely reflective of the situation in this municipality. This situation is not surprising, especially with regards to the Mankessim sub-district, as it hosts quite a large market which is heavily patronized. However, this was not initially anticipated and adjusted for.

The study focused only on short-acting hormonal contraceptives, that is the injectables and oral pills hence the findings of this study may not be applicable to other (long-acting) modern method users.

### Conclusion

The major thrust of this study was to assess factors associated with sustained use of modern contraceptives among women in the Mfantseman Municipality who continuously used their chosen methods despite the diverse challenges established for discontinuation. About 70% of the respondents had used their method for 2 or more years and almost all the respondents (97%) believed that the benefits of modern contraceptives outweighed the challenges. Some respondents, about 40%, reported socio-economic reasons such as saving money or getting a job before having a (next) child as their intention for their initial uptake of modern contraceptives. Among sustained users, almost all of them (99.6%) reported that their intent at initial adoption of modern contraceptives motivated their sustained use.

Again, the study revealed that age, privacy, limiting child-birth and ever missing a dose all had a statistically significant positive correlation with sustained use. Policymakers and healthcare practitioners should ensure that family planning environments are conducive for clients and that privacy for these clients are ensured.

Findings from the study suggest that it would be beneficial for providers to exercise greater tolerance towards women when they miss a dose of their method of contraceptive and provide the necessary support for them to resume use. Improved counseling after a missed dose is the probable cause of sustained use of contraceptive use among clients.

The finding that over two-thirds of the respondents had continued their methods for 2 or more years as well as the socio-economic importance attached to the use of modern contraceptives by these women is quite reassuring for family planning advocates. More so, almost all the participants reported challenges with use of their chosen method of contraception and yet continued use as they reckoned the advantages of modern contraceptive use. Reducing the incidence of unintended pregnancy through sustained use of modern contraceptives can therefore be enhanced with emphasis on benefits of use as well as ensuring privacy and confidentiality to clients. A multi-centre cohort study will give further insight into motivations for sustained contraceptive use.

## Supporting information

**S1 Data. Information such as raw data are available as supporting information.**
(XLSX)

## Acknowledgments

The authors are grateful healthcare workers of Mfantseman Municipality as well as research assistants and volunteers who contributed in diverse ways to the successful implementation of the study.

## Author contributions

**Conceptualization:** Seth Amponsah-Tabi, Hannah Boatemaa Asante, Charles Senaya, Roderick Larsen Reindorff, Henry S. Opare-Addo, Amponsah Peprah.

**Data curation:** Charles Senaya.

**Formal analysis:** Hannah Boatemaa Asante, Eric Sarpong Ansong, Henry S. Opare-Addo.

**Investigation:** Maxwell Kankam, Charles Senaya.

**Methodology:** Hannah Boatemaa Asante, Edward T. Dassah, Eric Sarpong Ansong, Henry S. Opare-Addo.

**Project administration:** Charles Senaya, Roderick Larsen Reindorff.

**Resources:** Amponsah Peprah.

**Software:** Eric Sarpong Ansong.

**Supervision:** Edward T. Dassah, Roderick Larsen Reindorff, Henry S. Opare-Addo.

**Validation:** Seth Amponsah-Tabi, Edward T. Dassah, Jude John Annan, Roderick Larsen Reindorff, Henry S. Opare-Addo.

**Visualization:** Edward T. Dassah, Jude John Annan, Roderick Larsen Reindorff, Amponsah Peprah.

**Writing – original draft:** Hannah Boatemaa Asante.

**Writing – review & editing:** Seth Amponsah-Tabi, Maxwell Kankam.

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
