## [Decision Letter · Decision Letter 0]

Dear Dr. Amponsah-Tabi,

Thank you for submitting your manuscript to PLOS ONE. After careful consideration, we feel that it has merit but does not fully meet PLOS ONE’s publication criteria as it currently stands. Therefore, we invite you to submit a revised version of the manuscript that addresses the points raised during the review process.

We look forward to receiving your revised manuscript.

Kind regards,

Surya Bahadur Parajuli, MD

Academic Editor

PLOS ONE

**Journal Requirements:**

3. Please include captions for your Supporting Information files at the end of your manuscript, and update any in-text citations to match accordingly. Please see our Supporting Information guidelines for more information: http://journals.plos.org/plosone/s/supporting-information .

4. We note that your Data Availability Statement is currently as follows: “All relevant data are within the manuscript and its supporting document”

Please confirm at this time whether or not your submission contains all raw data required to replicate the results of your study. Authors must share the “minimal data set” for their submission. PLOS defines the minimal data set to consist of the data required to replicate all study findings reported in the article, as well as related metadata and methods (https://journals.plos.org/plosone/s/data-availability#loc-minimal-data-set-definition ).

If your submission does not contain these data, please either upload them as Supporting Information files or deposit them to a stable, public repository and provide us with the relevant URLs, DOIs, or accession numbers. For a list of recommended repositories, please see https://journals.plos.org/plosone/s/recommended-repositories .

5. Please ensure that you refer all figures in your text as, if accepted, production will need this reference to link the reader to the figure.

**Reviewers' comments:**

Reviewer's Responses to Questions

**Comments to the Author**

1. Is the manuscript technically sound, and do the data support the conclusions?

Reviewer #1: Yes

2. Has the statistical analysis been performed appropriately and rigorously?

Reviewer #1: Yes

3. Have the authors made all data underlying the findings in their manuscript fully available?

Reviewer #1: Yes

4. Is the manuscript presented in an intelligible fashion and written in standard English?

Reviewer #1: Yes

Reviewer #1: Conciseness:

The abstract could be more concise. Aim to streamline sentences to focus on key findings without losing essential details.

Objective Clarity:

Clearly state the primary objective of the study at the beginning of the abstract. Instead of stating “this study sought to determine,” use “this study aimed to determine.”

Methodology Details:

Briefly mention the specific type of questionnaire used and any relevant details about its validity or reliability. This adds credibility to the methodology.

Statistical Data:

When presenting results, consider specifying the significance level (e.g., p-values) for the odds ratios (aORs) mentioned. This provides context for the strength of the associations.

Implications:

Strengthen the conclusion by emphasizing the practical implications of the findings for policymakers or health practitioners, perhaps by suggesting specific interventions.

Detailed Analysis:

Provide more detailed statistical analysis in the results. For example, include confidence intervals for the odds ratios to convey the precision of estimates.

Socio-Demographic Breakdown:

Expand on the socio-demographic section by providing a clearer picture of the sample. Consider including additional relevant variables (e.g., age distribution, income level) that might influence contraceptive use.

Current Use Information:

When discussing the current use of contraceptives, consider including reasons for choosing specific methods in a summarized format. This could be a table outlining key reasons for method choice.

Clarification of Terms:

Ensure that terms like "sustained users" are defined clearly when first introduced. This will help readers unfamiliar with the study context.

Subgroup Analysis:

Consider conducting and reporting subgroup analyses based on socio-demographic factors (e.g., age, education level) to understand how motivations for sustained use may differ across groups.

Motivations for Use:

Expand on the socio-economic motivations for sustained use. Include qualitative data or direct quotes from participants that highlight the diversity of motivations beyond statistics.

Discussion of Challenges:

While the results provide insight into motivations, consider briefly discussing the challenges respondents faced with their methods, as this could enrich the context.

Limitations of Findings:

Acknowledge any limitations directly in the results section, especially if there were potential biases or limitations in the sample selection.

Future Research Directions:

Suggest areas for future research based on the findings. This could help highlight gaps in knowledge that need to be addressed moving forward.

**Do you want your identity to be public for this peer review?** For information about this choice, including consent withdrawal, please see our Privacy Policy

Reviewer #1: No

---

## [Author Response · Author response to Decision Letter 1]

25 Jan 2025

PONE-D-24-29949

MOTIVATIONS FOR SUSTAINED USE OF MODERN CONTRACEPTIVES IN A PERI-URBAN AREA: AN ANALYTICAL CROSS-SECTIONAL STUDY.

PLOS ONE

3/01/25

RESPONSE TO REVIEWERS

Dear Sirs/Madam,

RESPONSE TO REVIEWERS

Thank you for responding to the manuscript. I am grateful for your review and I hope to meet all the requirements.

Thanks.

REVIEWER RESPONSE

1. The abstract could be more concise. Aim to streamline sentences to focus on key findings without losing essential details. RESPONSE: Changes have been made please.

2. Clearly state the primary objective of the study at the beginning of the abstract. Instead of stating “this study sought to determine,” use “this study aimed to determine.” RESPONSE: Changes have been effected.

3. Briefly mention the specific type of questionnaire used and any relevant details about its validity or reliability. This adds credibility to the methodology. RESPONSE: Researchers used open ended and closed questionnaire with face-to-face interviews.

4. When presenting results, consider specifying the significance level (e.g., p-values) for the odds ratios (aORs) mentioned. This provides context for the strength of the associations. RESPONSE: changes effected.

5. Strengthen the conclusion by emphasizing the practical implications of the findings for policymakers or health practitioners, perhaps by suggesting specific interventions. RESPONSE: Changes have been made to the conclusion please.

6. Provide more detailed statistical analysis in the results. For example, include confidence intervals for the odds ratios to convey the precision of estimates. RESPONSE: Changes effected please.

7. Expand on the socio-demographic section by providing a clearer picture of the sample. Consider including additional relevant variables (e.g., age distribution, income level) that might influence contraceptive use. RESPONSE: Thank you for drawing our attention to these other variables. Unfortunately, our initial data collection and analysis did not include these variables.

8. When discussing the current use of contraceptives, consider including reasons for choosing specific methods in a summarized format. This could be a table outlining key reasons for method choice. RESPONSE: Thanks for the input. The study however did not look into reasons why certain methods are preferred by our clients.

9. Ensure that terms like "sustained users" are defined clearly when first introduced. This will help readers unfamiliar with the study context. RESPONSE: Well noted.

10. Consider conducting and reporting subgroup analyses based on socio-demographic factors (e.g., age, education level) to understand how motivations for sustained use may differ across groups. RESPONSE: Thank you. We tried estimating the parameters for each of the age groups. Because each group (independent variable) was highly correlated, the estimates were statistically insignificant. They had p value of 1.000. as a result, we decided to treat age as a continuous variable to address the issue of multicollinearity.

11. Expand on the socio-economic motivations for sustained use. Include qualitative data or direct quotes from participants that highlight the diversity of motivations beyond statistics. RESPONSE: Thanks for drawing our attention. The study did not consider qualitative variables that might have an influence on sustained use.

12. While the results provide insight into motivations, consider briefly discussing the challenges respondents faced with their methods, as this could enrich the context. RESPONSE: Challenges respondents faced has been included. Line number 502)

13. Acknowledge any limitations directly in the results section, especially if there were potential biases or limitations in the sample selection. RESPONSE: has been noted.

14. Suggest areas for future research based on the findings. This could help highlight gaps in knowledge that need to be addressed moving forward. RESPONSE: Noted.

---

## [Decision Letter · Decision Letter 1]

Dear Dr. Amponsah-Tabi,

Thank you for submitting your manuscript to PLOS ONE. After careful consideration, we feel that it has merit but does not fully meet PLOS ONE’s publication criteria as it currently stands. Therefore, we invite you to submit a revised version of the manuscript that addresses the points raised during the review process.

We look forward to receiving your revised manuscript.

Kind regards,

Jianhong Zhou

Staff Editor

PLOS ONE

**Journal Requirements:**

Reviewers' comments:

Reviewer's Responses to Questions

**Comments to the Author**

Reviewer #2: (No Response)

2. Is the manuscript technically sound, and do the data support the conclusions?

Reviewer #2: Partly

3. Has the statistical analysis been performed appropriately and rigorously?

Reviewer #2: Yes

4. Have the authors made all data underlying the findings in their manuscript fully available?

Reviewer #2: Yes

5. Is the manuscript presented in an intelligible fashion and written in standard English?

Reviewer #2: Yes

**Reviewer #2:**  Thank you, the authors, for bringing this issue to the journal, as family planning is crucial for Africa and, particularly, highly populated countries like Nigeria. However, I have some concerns as follows:

- What is sustainability? As you have said, sustained. Do you think there is sustainability or continuity in using contraceptives?

- Where is the operational definition section also defining motivation, sustained, etc?

Measuring sustainability with a cross-sectional study is not proper enough. Have you matched the record from the health facility with the response from women? The authors should clarify this.

The result section doesn't clearly show what sustained users mean. Where is the operational definition? Who are new users?

Education and economy are the most important well-known factors, which are not new findings of this study. What are the unique findings specific to this manuscript?

**Do you want your identity to be public for this peer review?** For information about this choice, including consent withdrawal, please see our Privacy Policy

Reviewer #2: **Yes: ** Tizta T. Degfie

---

## [Author Response · Author response to Decision Letter 2]

9 May 2025

RESPONSE TO REVIEWERS

MOTIVATIONS FOR SUSTAINED CONTRACEPTIVE USE

7/5/25

Dear Editors,

Thank you for the recommendations and review questions.

RESPONSE TO REVIEWER

1. What is sustainability? As you have said, sustained. Do you think there is sustainability or continuity in using contraceptives?

RESPONSE: Sustained use is usage of a particular modern contraceptive by a client for 2 or more years.

2. - Where is the operational definition section also defining motivation, sustained, etc?

RESPONSE: Operational definition added to the methodology section; page 4.

3. Measuring sustainability with a cross-sectional study is not proper enough. Have you matched the record from the health facility with the response from women? The authors should clarify this

RESPONSE: We used both hospital records and response from participants for data gathering. Anomalies between hospital records and interview information were noted and excluded from the data. We have recommended a more advanced study in the conclusion to further determine sustainability of contraceptive use. Thank you.

4. The result section doesn't clearly show what sustained users mean. Where is the operational definition? Who are new users?

RESPONSE: Sustained users has been defined on page 11, line 262 to 263 at the results section. I have highlighted it to be easily identified.

New users have been included in the operational definitions on page 4; line 102-103. Thank you.

5. Education and economy are the most important well-known factors, which are not new findings of this study. What are the unique findings specific to this manuscript?

RESPONSE: Clients who experienced privacy in their counselling sessions for contraceptive uptake were more likely to have sustained use of a particular contraceptive method. Providing privacy and confidentiality is an important element in contraceptive counselling. However, this is not always available in resource constraint environments like the settings of this study. Clients who are limiters compared to spacers are also more likely to have sustained use of contraception as is a finding in this study. This can be found at the result section on page 20, line 402-405. Thank you.

---

## [Editor Report · Decision Letter 2]

MOTIVATIONS FOR SUSTAINED USE OF MODERN CONTRACEPTIVES IN A PERI-URBAN AREA: AN ANALYTICAL CROSS-SECTIONAL STUDY.

PONE-D-24-29949R2

Dear Dr. Amponsah-Tabi,

We’re pleased to inform you that your manuscript has been judged scientifically suitable for publication and will be formally accepted for publication once it meets all outstanding technical requirements.

Kind regards,

Christina M. Roberts, M.D., M.P.H.

Academic Editor

PLOS ONE
---

## [Editor Report · Acceptance letter]

PONE-D-24-29949R2

PLOS ONE

Dear Dr. Amponsah-Tabi,

I'm pleased to inform you that your manuscript has been deemed suitable for publication in PLOS ONE. Congratulations! Your manuscript is now being handed over to our production team.

Kind regards,

on behalf of

Dr. Christina M. Roberts

Academic Editor

PLOS ONE